# Geometric Exploitation for Indoor Panoramic Semantic Segmentation

**Duc Cao Dinh**          **Seok Joon Kim**          **Kyusung Cho**

Laboratory Department
MAXST
Seoul, Korea
`{caodinhduc, seokjoon, kscho}@maxst.com`

## Abstract

PAnoramic Semantic Segmentation (PASS) is an important task in computer vision, as it enables semantic understanding of a 360° environment. Currently, most of existing works have focused on addressing the distortion issues in 2D panoramic images without considering spatial properties of indoor scene. This restricts PASS methods in perceiving contextual attributes to deal with the ambiguity when working with monocular images. In this paper, we propose a novel approach for indoor panoramic semantic segmentation. Unlike previous works, we consider the panoramic image as a composition of segment groups: *over-sampled segments*, representing planar structures such as floors and ceilings, and *under-sampled segments*, representing other scene elements. To optimize each group, we first enhance *over-sampled segments* by jointly optimizing with a dense depth estimation task. Then, we introduce a *transformer-based context module* that aggregates different geometric representations of the scene, combined with a simple high-resolution branch, it serves as a robust hybrid decoder for estimating *under-sampled segments*, effectively preserving the resolution of predicted masks while leveraging various indoor geometric properties. Experimental results on both real-world (Stanford2D3DS, Matterport3D) and synthetic (Structured3D) datasets demonstrate the robustness of our framework, by setting new state-of-the-arts in almost evaluations, The code and updated results are available at: `https://github.com/caodinhduc/vertical_relative_distance`.

## 1   Introduction

In recent years, 360° camera images have garnered significant attention from learning systems and practical applications, including holistic sensing in automated vehicles [6, 8, 11, 14, 17, 20] and immersive experiences in augmented reality (AR) and virtual reality (VR) devices [1, 29]. Unlike traditional pinhole cameras with their limited Fields of View (FoV), panoramic images offer an expansive 360° × 180° FoV, providing a more comprehensive perception of both indoor and outdoor environments. This wide FoV enhances numerous fundamental computer vision tasks by enabling a richer understanding of scenes, thus benefiting many fundamental computer vision tasks. One such task, Panoramic Semantic Segmentation (PASS) is a pivotal task that generates dense, pixel-wise class maps, significantly improving high-level understanding of complex environments. By harnessing the wide field of view and unique properties of panoramic images, PASS enables more comprehensive scene analysis, delivering valuable insights across a range of applications.

Most current Panoramic Semantic Segmentation (PASS) approaches rely on 2D panoramas transformed through equirectangular projection [24, 33]. However, these methods face two major challenges: limited annotated data and significant image distortions. In terms of data scarcity, existing

38th Conference on Neural Information Processing Systems (NeurIPS 2024).

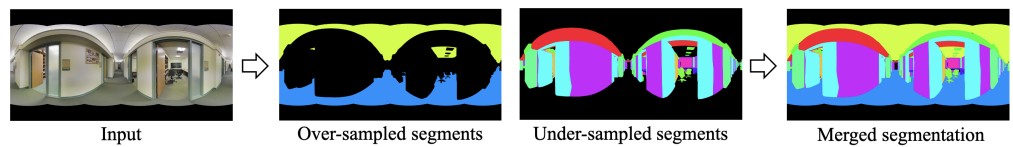

| Input | Over-sampled segments | Under-sampled segments | Merged segmentation |

Figure 1: Panoramic segmentation problem can be re-formulated to the estimation of *over-sampled segments:* floor, ceiling and *under-sampled segments*: chair, table, bookcase, window, etc.

datasets are small and lack scene diversity due to the labor-intensive process of manually labeling and verifying segments in each image. For example, the widely-used Stanford 2D3DS dataset [2] contains only 1,413 multi-modal equirectangular images spanning 13 object categories, which is insufficient for robust deep learning model development—posing a substantial challenge for PASS tasks. To address this challenge, aside from unsupervised domain adaptation approaches [19, 35, 38], which are effective but predominantly applied to outdoor scenes, the development of a large-scale synthetic dataset [37], featuring detailed 3D structural annotations and photo-realistic 2D renderings of indoor environments, presents a promising solution. As for the second issue, the conversion of panoramic images from spherical to rectangular coordinates results in the oversampling of regions near the poles (floor and ceiling) compared to those near the equator in the original 360° data. This distortion leads to an imbalance in the relative sizes of different classes within each image. Supporting this observation, a quantitative analysis of the Stanford2D3DS dataset shows that, on average, ceilings and floors occupy 18% and 20% of a panoramic image, respectively. Consequently, larger classes such as floors and ceilings are easier to predict, while smaller, more intricate classes like chairs and tables pose greater challenges, as reflected in the disparity of quantitative results between these groups.

In this work, to tailor optimization for different regions of panoramic images more effectively, instead of designing a deep learning network which performs to entire image, we divide the equirectangular RGB input into two subgroups of segments. As shown in the Figure 1, we reformulate the problem of panoramic semantic segmentation as the estimation of *over-sampled segments* (floor and ceiling) and *under-sampled segments* (chair, table, window, etc.). We then exploit indoor scene geometric properties to tailor optimization for each group with different strategies as follows: 1) We propose a collaborative study on semantic segmentation of the *over-sampled segments* and dense depth estimation, this approach enables each task to leverage the benefits of the other, not only through implicit cross-task representation but also by enforcing consistent geometric losses. 2) To fully leverage the rich geometric information in panoramic images, we go beyond the common approaches to introduce a novel concept of *vertical relative distance* which indicates the relative positions of 3D points with respect to the key components of indoor scenes. These relative distances, combined with image features and other geometric representations are incorporated by a designated *transformer-based context module*, along with a simple high resolution branch, it can be served as a hybrid decoder optimized for the *under-sampled segments* estimation.

In this paper, we present a novel approach to Indoor Panoramic Semantic Segmentation (PASS). Our key contributions are as follows:

- We propose a new method for PASS that decomposes the task into sub-problems and optimizes them by integrating geometric information through distinct strategies.
- We introduce the *vertical relative distance*, a new geometric representation that captures the spatial relationships between planar surfaces (ceilings and floors) and other object pixels in 3D space.
- We design a hybrid decoder combining a simple high-resolution branch with a *transformer-based context module*, which integrates scene representations and exploits relationships among geometric components.
- Our framework achieves state-of-the-art performance, demonstrating robustness, accuracy, and efficiency on publicly available panoramic semantic segmentation datasets.

To evaluate the effectiveness of our proposed techniques, we benchmark our framework against baseline models and prior methods on three widely-used panoramic semantic segmentation datasets: the real-world Stanford2D3DS, Matterport3D, and the synthetic Structured3D. On the Stanford2D3DS evaluation (fold 1), our method achieves a new state-of-the-art performance with 56.8% mIoU. On

the Matterport3D dataset, we surpass previous methods under the same input conditions, reaching 33.06% mIoU on testing set. A similar trend is observed on the Structured3D test set, where our model attains 71.66% mIoU, demonstrating its robustness across diverse scenarios.

## 2 Related Work

**Panoramic Semantic Segmentation.** With recent advances in deep learning, numerous neural network-based methods have emerged for panoramic semantic segmentation. Deng et al. [9] were among the first to convert a pinhole urban traffic scene dataset into wide-angle (fisheye) images, introducing a pioneering framework for panoramic semantic segmentation. Yang et al. [32] later proposed a method for semantic segmentation on 360-degree panoramic annular images, captured with a single panoramic camera, to improve full-field environmental perception. Building on this, they developed DS-PASS [31], which enhances their earlier work by incorporating attentional mechanisms to improve efficiency in panoramic segmentation.

To address distortion in equirectangular images, Tateno et al. [25] introduced distortion-aware convolutions, where the convolutional filter adapts its shape based on the level of distortion in the projected image. Similarly, Zhuang et al. [39] proposed Adaptively Combined Dilated Convolution (ACDNet), which enhances the Field of View near the poles of spherical images by using dilated convolutions as a direct replacement for standard convolutions. Su and Grauman [23] introduced Spherical Convolution, a technique that adjusts kernel sizes dynamically based on spherical coordinates. Coors et al. [7] developed SphereNet, which handles Equirectangular Projection (ERP) by modifying the sampling grid positions of convolution filters to achieve distortion invariance, allowing end-to-end training. Along the same lines, Zhao et al. [36] proposed a distortion-aware CNN for 360-degree spherical images, incorporating both distortion-aware convolutional and pooling layers. Khasanova and Frossard [15] took a different approach by replacing traditional convolutional filters with graph-based filters that adjust dynamically according to the position within the omnidirectional image. Finally, Jiang et al. [13] introduced a novel convolution kernel for CNNs on arbitrary manifolds and topologies, utilizing parameterized differential operators discretized via an unstructured mesh.

Sharing a similar vision but employing different approaches, Zhang et al. [34, 35] introduced Trans4PASS and Trans4PASS+, which address spherical distortions and image deformations using Deformable Patch Embedding (DPE) and Deformable Multi-Layer Perception (DMLP) modules. Li et al. [18] proposed a Spherical-Geometry-Aware method to handle distortions when converting 360-degree data into 2D panoramic images. Building on the strengths of DPE and DMLP, we adopt Trans4PASS+ as the baseline model for this paper.

**Multi-task approach for the Panoramic Semantic Segmentation.** A Panoramic Semantic Segmentation network can be trained jointly with other computer vision tasks to leverage additional 3D geometric information from the scene, helping to resolve ambiguities that are challenging for purely 2D approaches. HoHoNet [24] introduced a framework for jointly predicting layout structure and performing dense per-pixel tasks, such as depth estimation and semantic segmentation, based on a 1D horizontal feature representation. Similarly, Berenguel-Baeta et al. [4] proposed a method to jointly perform semantic segmentation and depth estimation from a single equirectangular panorama, utilizing Fourier convolution (FFC) to expand the receptive field and enhance feature extraction. Following a similar intuition, MultiPanoWise [21] extends vision transformers to jointly infer multiple pixel-wise estimation tasks along with signals from intrinsic decomposition.

## 3 Method

This section details the architecture of our proposed network, a novel approach for panoramic semantic segmentation from an RGB equirectangular image. As illustrated in the Figure 2, the network consists of three key modules: (1) an encoder that generates both high-resolution coarse features and low-resolution fine features, (2) a branch dedicated to the concurrent estimation of the *over-sampled segments* (floor and ceiling) along with dense depth estimation, and (3) a hybrid decoder that integrates a deformable MLP with a novel *transformer-based context module* to produce semantic masks for the *under-sampled segments*.

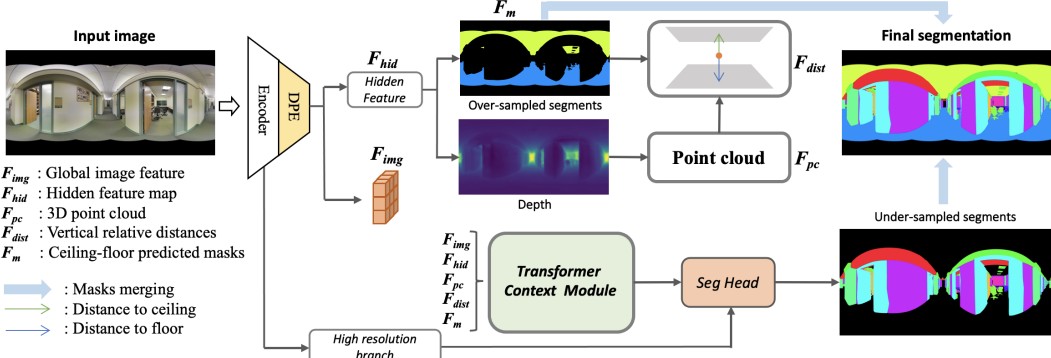

Figure 2: The proposed framework consists of three main modules: an encoder for extracting image features, a branch that estimates *over-sampled segments* alongside dense depth estimation, and a hybrid decoder for estimating *under-sampled segments* before a merging process to obtain the final segmentation result.

## 3.1 Encoder

Given an input image of size $H \times W \times 3$, we adopt the encoder architecture from Trans4PASS+ [35] to generate multi-level feature maps, denoted as $f = \{f_1, f_2, f_3, f_4\}$. These feature maps correspond to resolutions of $\{1/4, 1/8, 1/16, 1/32\}$ of the original image size, with channel dimensions $C = \{64, 128, 320, 512\}$, respectively. Following the approach from the baseline [35], we apply Deformable Patch Embedding (DPE) modules to the image features at each level before the decoding stage. This transforms the feature hierarchy into embedded feature maps, represented as $e = \{e_1, e_2, e_3, e_4\}$, while maintaining hierarchical resolution, the channel dimensions are unified to $C_{emb}$. In our network, we set the number of embedding channels to $C_{emb} = 128$.

## 3.2 Over-sampled segments estimation

We design a simple decoder to jointly estimate dense depth and *over-sampled segments*. Given the embedded feature maps $e = e_1, e_2, e_3, e_4$, we apply deformable MLPs at each level and upscale the spatial resolution to $\frac{H}{4} \times \frac{W}{4}$. These upscaled features are then fused using point-wise summation to obtain a hidden feature map ($F_{hid}$) of size $\frac{H}{4} \times \frac{W}{4} \times C_{emb}$.

Next, lightweight prediction heads process the hidden features to estimate both dense depth ($\frac{H}{4} \times \frac{W}{4} \times 1$) and over-sampled segments ($\frac{H}{4} \times \frac{W}{4} \times N_{cls}$), where $N_{cls} = 3$, corresponding to *floor, ceiling,* and *unknown*, represents the union of all other classes. Unlike regular objects, ceilings and floors are typically planar surfaces in 3D environments.

To preserve these planar properties, we leverage the Plane Surface Normal Loss (PSN) from Xie et al. [27] to enforces geometric constraints on the predicted depth map by utilizing ground truth floor-ceiling masks. This facilitates accurate plane equation fitting in subsequent processing steps.

## 3.3 Contextual Information Exploitation

After obtaining the dense depth for the entire image and the predicted floor-ceiling masks, we introduce several approaches to extract indoor geometric representations. These representations, combined with the global image feature map, form the input for the proposed *transformer-based context module*, which enhances the scene understanding for the *under-sampled segments* estimator.

**Global Image Feature.** We consider the embedded feature map $e = e_1, e_2, e_3, e_4$ to form the global feature of image. To streamline the patch embedding process in the *transformer-based context module*, we upsample low-resolution embedded feature maps ($e_2, e_3, e_4$), then stack them with $e_1$ along the channel dimension to create the global image feature denoted as $F_{img}(\frac{H}{4} \times \frac{W}{4} \times 4 * c_{emb})$.

**Cross-Task Feature Distillation.** An intuitive way to help the segmentation model understand scene context is to distill representations from the hidden features of other tasks within a multi-task network,

as proven by the effectiveness shown in [4, 28, 30]. In this work, we propose distilling information from the hidden feature map $F_{hid}$ ($\frac{H}{4} \times \frac{W}{4} \times C_{emb}$), introduced in Section 3.2. This map not only captures early depth features but also implicitly encodes the estimation of floor-ceiling planes.

**3D Point Cloud.** Another approach to enhance the segmentation estimator's awareness of 3D context is by first predicting a depth map from the input RGB image and incorporating it as input to the segmentation network. However, we argue that using point cloud data in Cartesian coordinates offers a more robust geometric understanding than depth maps. Point clouds represent 3D data directly, making them better suited for capturing the spatial layout and geometry of a scene. Specifically, given the predicted depth map as 3.2 at resolution of $H \times W$, we apply an Equirectangular to Cartesian projection (detail in appendix) to generate a point cloud set $S \in \mathbb{R}^{N \times 3}$, where $N = H \times W$. We then resize and format this point cloud into $F_{pc}$ ($\frac{H}{4} \times \frac{W}{4} \times 3$) for further processing.

**Vertical Relative Distance.** Both cross-task feature distillation and 3D point clouds efficiently capture geometric information about the surrounding environment, but neither fully leverages the distinctive characteristics of indoor scenes (e.g., chairs typically rest on the floor, while lights hang from the ceiling). As a key observation, floors and ceilings are prominent features that dominate most indoor panoramic images, acting as critical constraints in these scenes. In 3D coordinates, the floor and ceiling usually form parallel planes, creating the upper and lower boundaries of the space. Thus, the total distance between a point to the floor and ceiling tends to remain constant.

These distances not only reflect a point's position relative to the floor-ceiling planes but also describe its spatial relationship to the key constraints derived from the panoramic image. Based on this observation, we introduce a novel concept called *vertical relative distance*, which utilizes prior information such as point cloud data ($S \in \mathbb{R}^{N \times 3}$) and predicted floor-ceiling masks. This new metric is constructed as follows:

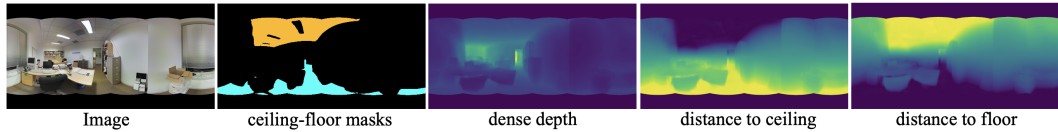

| Image | ceiling-floor masks | dense depth | distance to ceiling | distance to floor |

Figure 3: Visualization of an example showing *distance to floor* and *distance to ceiling* masks. The gradient from light to dark represents the transition from greater to shorter distances.

We define $S_f \in \mathbb{R}^{N_f \times 3}$ and $S_c \in \mathbb{R}^{N_c \times 3}$ as the sets of floor and ceiling point clouds, respectively. For the floor set $S_f$, the equation of the floor plane $e_f$ in Cartesian coordinates is given by:

$$a_f x + b_f y + c_f z + d_f = 0 \tag{1}$$

We apply the least squares method (detail in appendix) to $S_f$ to determine the plane coefficients $a_f, b_f, c_f$, and $d_f$. The distance from each point $p(x, y, z) \in S$ to the floor plane $e_f$ is then calculated as follows:

$$d(p, e_f) = \frac{|a_f x + b_f y + c_f z + d_f|}{\sqrt{a_f^2 + b_f^2 + c_f^2}} \tag{2}$$

By applying this process to the entire set $S$, we obtain $d_f \in \mathbb{R}^{H \times W}$, representing the distances of points in $S$ to the floor plane. Similarly, using this procedure, we compute the distances of points in $S$ to the ceiling plane, denoted as $d_c \in \mathbb{R}^{H \times W}$. The resulting pair of tensors, $d_f$ and $d_c$, are stacked and resized to form $F_{dist} \in \mathbb{R}^{\frac{H}{4} \times \frac{W}{4} \times 2}$, which encodes the *vertical relative distance* of the indoor scene (as shown in Figure 3). Additionally, we leverage the predicted floor-ceiling mask $F_m \in \mathbb{R}^{\frac{H}{4} \times \frac{W}{4} \times 2}$, which provides prior positional information, highlighting the precise locations of the constraint components within the panoramic images.

### 3.4 Under-sampled segments estimation

While estimating *over-sampled segments* is relatively straightforward, predicting *under-sampled segments* poses a greater challenge due to the dense appearance of objects at varying scales and the inherent ambiguities of working with monocular images. Our goal is to design a robust decoder that efficiently leverages image features while capturing the intrinsic relationships between contextual

and geometric elements within the indoor scene. To achieve this, we propose a hybrid decoder with the following structure:

**Transformer-based Context Module.** Given a single panoramic image, our objective is to improve the extraction of representations that capture both global image features and diverse geometric structures. As illustrated in the Figure.4, the global image feature $F_{img}$ is combined with the upper branch hidden feature map $F_{hid}$, 3D point cloud $F_{pc}$, vertical relative distance $F_{dist}$, and prior predicted floor-ceiling masks $F_m$. These concatenated features serve as the input to the context module:

$$Z = [F_{img}, F_{hid}, F_{pc}, F_{dist}, F_m] \tag{3}$$

The context module is composed of patch and positional embeddings, followed by six stacked transformer encoder layers [26]. Each layer consists of a multi-head self-attention (MHSA) mechanism and a feed-forward network (FFN). MHSA, a key component of the transformer architecture, allows the model to simultaneously attend to information from multiple representation subspaces. In the self-attention module, the input embedding $Z$ is passed through three projection matrices ($W^Q$, $W^K$, $W^V$) to generate the query ($Q$), key ($K$), and value ($V$) embeddings.

$$Q = ZW^Q, K = ZW^K, V = ZW^V \tag{4}$$

The output of self-attention is the aggregation of the values that are weighted by the attention weights:

$$SA(Q, K, V) = softmax(\frac{QK^T}{\sqrt{d}})V \tag{5}$$

Where $d$ is the dimension of query embedding. Multiple self-attention layers are stacked and their concatenated outputs are fused by weighting matrix $W^h$, to form MHSA:

$$MHSA(Q, K, V) = \sum_{h=1}^{H} MSA(Q, K, V)W^h \tag{6}$$

After exploiting the relationship between global image feature with different geometric information through a sequence of transformer encoder layers, the output tokens can be reshaped into an image-like and upsampled to a size of $\frac{H}{4} \times \frac{W}{4}$ by strided $3 \times 3$ transpose convolution for the next processing.

**Combine with a high-resolution branch.** While transformer encoder layers are effective at capturing long-range dependencies, they typically produce low-resolution outputs ($\frac{H}{16} \times \frac{W}{16}$ before upscaling in our case), which hinders the decoder's ability to generate high-resolution, dense predictions. To address this limitation, we introduce a high-resolution branch that directly extracts image features $c1$ by applying a Deformable MLP module, generating features at a resolution of $\frac{H}{4} \times \frac{W}{4}$. These features are then fused with the context module output via element-wise summation. Finally, a lightweight head is employed to accurately estimate fine-grained masks for *under-sampled segments* at a resolution of $\frac{H}{4} \times \frac{W}{4} \times N_{cls}$, where $N_{cls}$ represents the number of classes for under-sampled segments, with an additional class for the *unknown*.

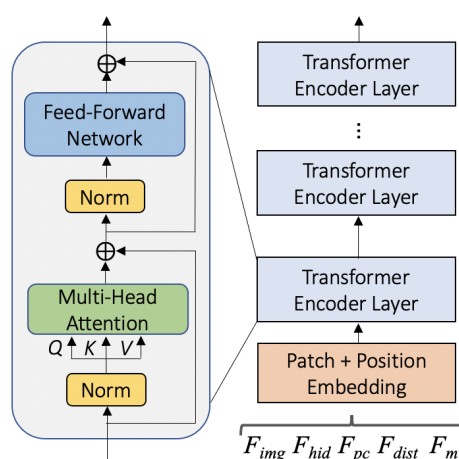

Figure 4: Transformer based context module

### 3.5 Merged segments estimation

Since the over-sampled and under-sampled segments are estimated independently, a straightforward merging mechanism combines them to produce the final segmentation for the entire image. First, the two sets of predicted masks are unified. For overlapping segments, priority is given to floor or ceiling predictions, as they tend to be more accurate and stable. Finally, the merged segments is upscaled to match the original image resolution.

# 4 Experiments

## 4.1 Experiment Settings

**Datasets:** We utilize three publicly available datasets for comparison: Stanford2D3DS [3], Structured3D [37], and Matterport3D [5]. The Stanford2D3DS dataset contains 1,413 real-world panoramic images with annotations for 13 semantic classes, organized into three official folds. We follow the fold-splitting scheme established in previous works [4, 18, 34, 35]. The Structured3D dataset, on the other hand, provides 21,835 synthetic equirectangular images with diverse lighting setups and annotations for 40 semantic classes. Consistent with [12, 37], we define training, validation, and test splits as follows: scenes 00000–02999 for training, scenes 03000–03249 for validation, and scenes 03250–03499 for testing. For all evaluations, we use raw rendered images under full lighting and furniture configurations. Meanwhile, the Matterport3D dataset introduces an additional challenge with its 10,800 panoramic images capturing 40 semantic classes in complex indoor scenes. Following the processing and split protocol of Guttikonda and Rambach [12], we create training, validation, and test subsets for consistency in our experiments.

**Implementation details** We train our model on a single NVIDIA GeForce RTX 3090 GPU, starting with an initial learning rate of 5e-5, adjusted using a poly decay strategy with a power of 0.9 over the training epochs. For the Stanford2D3DS, Structured3D, and Matterport3D datasets, we train for 100, 50, and 100 epochs, respectively. The AdamW optimizer [16] is used with an epsilon of 1e-8, a weight decay of 1e-4, and a batch size of 4. Image augmentations include random horizontal flipping, random cropping, and resizing to $512 \times 1024$. In the testing phase, images are also processed at a resolution of $512 \times 1024$. Other settings and hyperparameters match those of Tran4PASS+ [35]. For the segmentation and depth estimation tasks, we use Focal and Huber losses, respectively, with the final training loss computed as a combination:

$$L_{total} = \alpha_1 . L_{over-sampled-segment} + \alpha_2 . L_{under-sampled-segment} + \alpha_3 . L_{depth} \tag{7}$$

Here, $L_{over-sampled-segment}$ and $L_{under-sampled-segment}$ represent the segmentation losses for the *over-sampled* and *under-sampled* segments estimation, respectively. In our experiments, the weights $\alpha_1$, $\alpha_2$, and $\alpha_3$ are set to [1, 5, 1].

## 4.2 Experiment Results

Table 1: Comparison with state-of-the-art methods on the Stanford2D3DS dataset. Consistent with recent work, we report performance as the average mIoU across all three official folds (Avg mIoU) and on fold 1 specifically (F1 mIoU). Our approach demonstrates substantial improvements over both the baseline and existing methods.

| Method | Venue | Validation Avg mIoU (%) | Validation F1 mIoU (%) |
|---|---|---|---|
| Tangent [10] | CVPR 2019 | 45.6 | - |
| FreDSNet [4] | ICRA 2022 | - | 46.1 |
| PanoFormer [22] | ECCV 2022 | 48.9 | - |
| SFSS-MMSI [12] | WACV 2024 | - | 52.9 |
| HoHoNet [24] | CVPR 2021 | 52.0 | 53.9 |
| Trans4PASS [34] | CVPR 2022 | 52.1 | 53.3 |
| Trans4PASS+ [35] | Arxiv 2022 | 53.7 | 53.6 |
| SGAT4PASS [18] | IJCAI 2023 | 55.3 | 56.4 |
| **Ours** | | **55.5** | **56.8** |

**Results on the Stanford2D3DS dataset:** Table 1 presents a quantitative comparison of our framework with various panoramic semantic segmentation methods on the Stanford2D3DS validation and test sets. Our method demonstrates significant robustness, achieving a 1.8% and 3.1% mIoU improvement over the baseline Trans4PASS+, respectively, surpassing the current state-of-the-art performance by a small margin. The qualitative results in the Figure 5 further report the advantages of our approach in handling diverse geometric representations. Our model consistently outperforms previous methods, particularly in regions where similar RGB features lead to confusion. For instance, in the first row, the board and wall share similar RGB colors, making it challenging for previous methods to differentiate between them. In contrast, our method accurately distinguishes these classes. Overall, prior methods

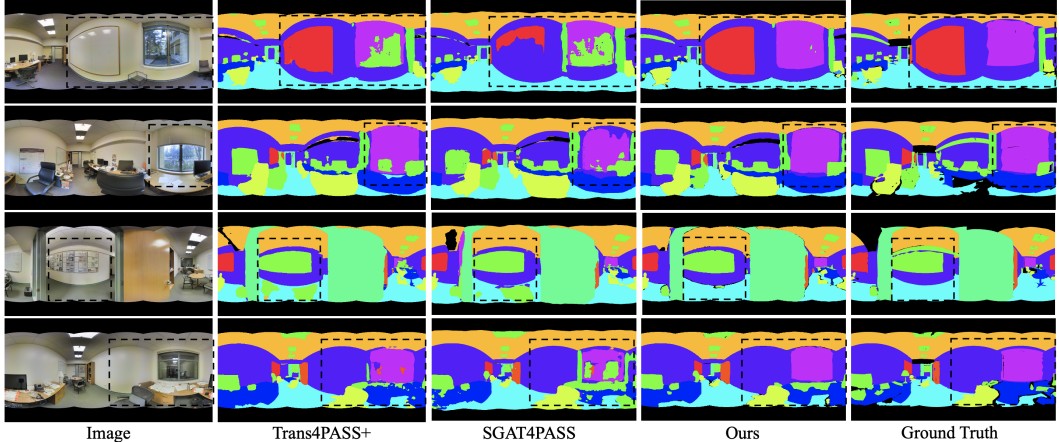

| Image | Trans4PASS+ | SGAT4PASS | Ours | Ground Truth |

Figure 5: Qualitative comparison of semantic segmentation results from Trans4PASS+ [35], SGAT4PASS [18], and ours using the Stanford2D3D dataset. Black boxes highlight the improvements. Zoom for the better view.

that estimate all segments using a single network tend to yield results optimized for large segments (such as the floor and ceiling) but struggle with segments near the image equator. In contrast, our proposed learning strategy mitigates the effects of imbalanced distortion across indoor panoramic images, resulting in clearer estimations of smaller segments such as chairs, boards, and tables.

Table 2: Quantitative comparison of depth estimation task.

| Method | MRE↓ | MAE↓ | RMSE↓ | RMSElog↓ | $\delta_1 \uparrow$ | $\delta_2 \uparrow$ | $\delta_3 \uparrow$ |
|---|---|---|---|---|---|---|---|
| FreDSNet [35] | 0.095 | 0.133 | 0.518 | 0.208 | 0.843 | 0.958 | 0.986 |
| **Ours** | **0.074** | **0.120** | **0.390** | **0.760** | **0.865** | **0.988** | **0.991** |

**Performance of the depth estimation task on the Stanford2D3DS dataset:** Since our work integrates depth estimation within the network, we also report the performance of the depth estimation task. We compare our framework to FreDSNet [4], a multi-task learning model for joint panoramic semantic segmentation and depth estimation. The evaluation metrics include Mean Relative Error (MRE), Mean Absolute Error (MAE), Root-Mean Square Error (RMSE), logarithmic RMSE (RMSElog), and three relative accuracy measures defined as the fraction of pixels with a relative error within thresholds of $1.25^n$ for n = 1,2,3 ($\delta 1, \delta 2, \delta 3$). As result shown in the Table 2, our network outperforms FreDSNet at all of quantitative metrics, demonstrate the powerfulness of our framework not only on segmentation but also depth estimation.

**Results on the Structured3D dataset:** We further conduct the experiment on Structured3D synthetic dataset, which provides a greater variety of images and classes. As the illustration in the Table 3, on both validation and test sets, our method proves the effectiveness, outperforms previous works to mark a new state of the art performance with 72.86% mIoU and 71.66% mIoU, respectively. Qualitative comparisons in the Figure 6 highlight the robustness of our framework.

**Results on the Matterport3D dataset:** Lastly, we evaluate the proposed approach on the challenging Matterport3D dataset, which features diverse classes within complex indoor scenes. As shown in Table 3, given only RGB input, our network slightly outperforms previous methods, achieving new peaks with quantitative results of 36.42% mIoU on the validation set and 33.06% mIoU on the test set. These metrics are competitive with methods that utilize additional dense depth (RGBD) input. However, due to the inherent difficulty of the Matterport3D dataset, the results across all methods are generally constrained, as illustrated in the qualitative visualization in Figure 7.

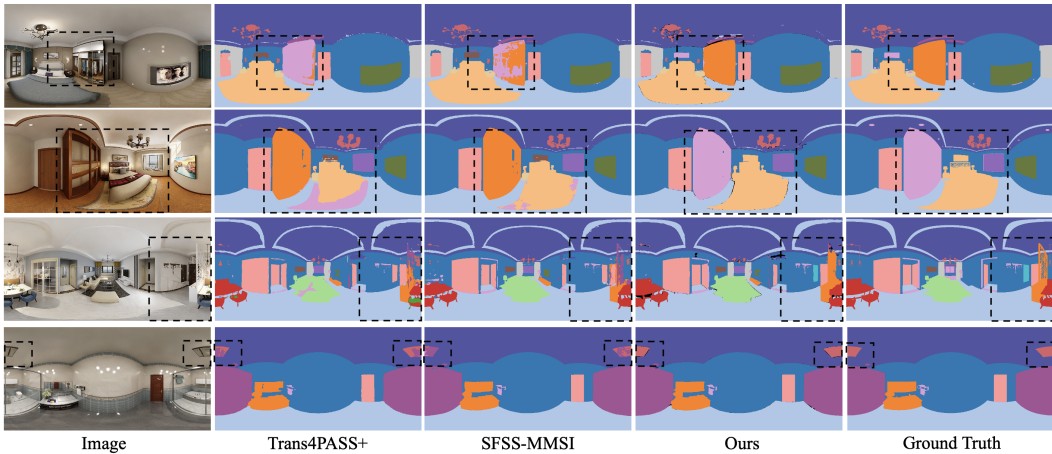

| Image | Trans4PASS+ | SFSS-MMSI | Ours | Ground Truth |

Figure 6: Qualitative comparison of semantic segmentation results from Trans4PASS+ [35], SFSS-MMSI [12], and ours using the Structured3D dataset. Black boxes highlight the improvements.

Table 3: Comparison with the SOTA methods on the Structured3D and the Matterport3D validation and test sets. Our method marks new state of the arts on both datasets given the same input.

| Methods | Modal | Structured3D | | Matterport3D | |
| | | Val mIoU (%) | Test mIoU (%) | Val mIoU (%) | Test mIoU (%) |
| --- | --- | --- | --- | --- | --- |
| PanoFormer [22] | RGB | 55.57 | 54.87 | 30.04 | 26.87 |
| Trans4PASS+ [35] | RGB | 66.74 | 66.90 | 33.43 | 29.21 |
| SFSS-MMSI [12] | RGB | 71.94 | 68.34 | 35.15 | 31.30 |
| PanoFormer [22] | RGBD | 60.98 | 59.27 | 33.99 | 31.23 |
| SFSS-MMSI [12] | RGBD | **73.78** | 70.17 | **39.19** | **35.92** |
| **Ours** | RGB | **72.86** | **71.66** | **36.42** | **33.06** |

### 4.3 Ablation study

**Impact of each contextual representation.** We conduct the ablation study on the Stanford2D3DS dataset [3] (fold 1) to measure the influence of different geometric representation to the performance of network. We consider a baseline setting with global image feature is input of *transformer-based context module*. After that, model is train with additional geometric properties as input of the *transformer-based context module*, detail of the performance is reported in the Table 4.

Table 4: Impact of each geometric representation to the model performance.

| Geometric properties | mIoU (%) | Pixel Acc (%) |
| --- | --- | --- |
| $F_{img}$ | 54.41 | 82.06 |
| $F_{img} + F_{hid}$ | 54.83 | 82.46 |
| $F_{img} + F_{hid} + F_{pc}$ | 55.32 | 82.87 |
| $F_{img} + F_{hid} + F_{pc} + F_{dist}$ | 56.68 | 83.32 |
| $F_{img} + F_{hid} + F_{pc} + F_{dist} + F_m$ | **56.80** | **83.45** |

**Model performance with and without the involvement of depth.** We also conducted ablation studies on the Stanford2D3DS dataset (fold 1) with the baseline( [35]), our method, and additional two configurations to analyze the role of depth estimation task. In the first setting, we retained the segment partitioning and optimization strategy but removed both the depth branch and the features $F_{pc}$ and $F_{dist}$ from the input of the *transformer-based context module*. In the second setting, we maintained joint learning for over-sampled segments with depth estimation, but excluded $F_{pc}$ and $F_{dist}$ from the *transformer-based context module*, detail of the quantitative comparison is shown in the Table 5.

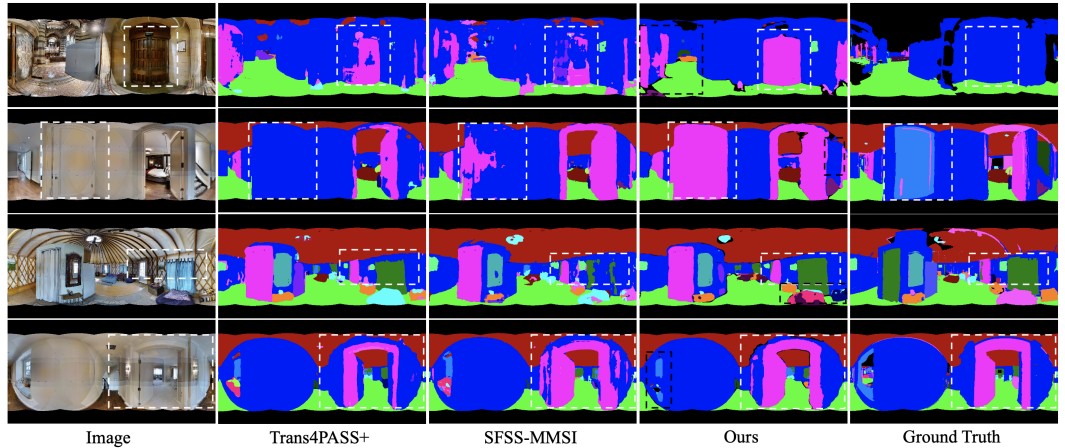

| Image | Trans4PASS+ | SFSS-MMSI | Ours | Ground Truth |

Figure 7: Qualitative comparison of semantic segmentation results from Trans4PASS+ [35], SFSS-MMSI [12], and ours using the Matterport3D dataset. Our method generally performs better than previous approaches (highlighted in white boxes). However, some challenges remain in regions with ambiguous textures (highlighted in black boxes). Zoom in for better view.

Table 5: Impact of depth estimation involvement to the model performance.

| Settings | Avg mIoU (%) | Fold 1 mIoU (%)) |
|---|---|---|
| Baseline | 53.7 | 53.6 |
| Without depth involvement (depth estimation, both $F_{pc}$ and $F_{dist}$) | 54.3 | 54.7 |
| With depth estimation but no $F_{pc}$ and $F_{dist}$ | 54.6 | 55.0 |
| With full depth involvement | **55.5** | **56.8** |

**Model complexity.** Table 6 presents the model complexity, comparing the number of parameters and TFLOPs of our approach against previous methods.

Table 6: Computational complexity comparison with input size: $512 \times 1024 \times 3$.

| Input | Params(G) | TFLOPs |
|---|---|---|
| Trans4PASS+ [35] | 0.039 | 0.131 |
| HoHoNet [24] | 0.070 | 0.125 |
| SFSS-MMSI [12] | 0.040 | 0.079 |
| PanoFormer [22] | 0.020 | 0.081 |
| Ours | 0.053 | 0.135 |

## 5   Conclusion

In this paper, we introduce a novel approach for Indoor Panoramic Semantic Segmentation that decomposes the task into sub-problems, optimizing each by leveraging geometric information in distinct ways. We treat the floor and ceiling as constrained components of the panoramic image and propose *vertical relative distance* as a new geometric representation of the indoor scene. Additionally, we design a hybrid decoder with a *transformer-based context module* to aggregate diverse geometric properties effectively. Our framework demonstrates both efficiency and robustness, achieving new state-of-the-art performance on three public datasets. Despite its strong performance, our method has some limitations. First, since the two groups of segments are estimated separately, some *unknown* areas can emerge during the merging process. Second, the addition of a depth estimation branch increases the model's complexity considerably (as report in the Table 6). Besides, the performance on the Matterport3D dataset remains limited. Future work should focus on exploring robust methods that incorporate a deeper understanding of scene structure to further enhance the panoramic semantic segmentation task.

## Acknowledgments and Disclosure of Funding

We thank MAXST Co. Ltd for their support in providing time and infrastructure for this project.

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

# A  Appendix

We provide the appendix, including the detailed algorithm explanations from the main paper, color bars for segmentation labels, examples of distance to ceiling, floor masks, and 2D/3D visualizations of our approach. Additional qualitative/quantitative results will be updated on github, please follow the repository at: `https://github.com/caodinhduc/vertical_relative_distance`.

## A.1  Projection of equiretangular image to 3D points in Cartesian coordinates using depths.

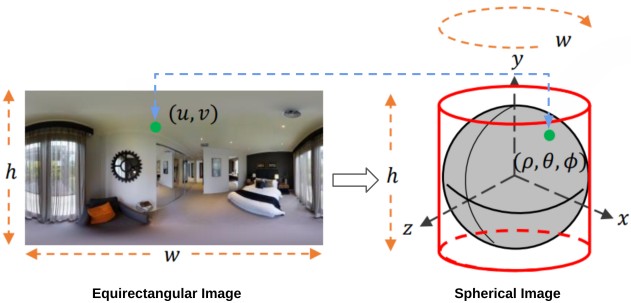

Figure 8: Convert equirectangular image to spherical image. Image is adjusted from Ai et al. [1].

A pixel from equirectangular image can be projected to spherical coordinate, then be converted to Cartesian coordinates $(x, y, z)$ using a depth map estimated as described in Section 3.2. Pixels in equirectangular images can be defined: $u, v$, where $u$ ranges from $0$ to $w$ and $v$ ranges from $0$ to $h$. If we define the horizontal unit angle as $\vartheta = \frac{2\pi}{w}$ and the vertical unit angle as $\varphi = \frac{\pi}{h}$, each pixel $(u, v)$ in the ERP can be mapped to the spherical coordinate $(\theta, \phi)$ as shown in Figure 8:

$$(\theta, \phi) = (u \cdot \vartheta, v \cdot \varphi)$$

Here, $\rho$ is the radius or depth, $\theta \in [0, 2\pi]$ is the longitude angle, and $\phi \in [0, \pi]$ is the latitude angle.

To convert from spherical coordinates $(\theta, \phi)$ to Cartesian coordinates $(x, y, z)$ using depth maps, the following formulas are used: $x = \rho \sin(\theta) \sin(\phi)$, $y = \rho \cos(\phi)$, and $z = \rho \cos(\theta) \sin(\phi)$.

## A.2  Least Square Method for plane fitting from point cloud.

The least squares method is a statistical technique used to find the best-fitting plane to a set of three-dimensional points $(x_i, y_i, z_i)$. The equation of a plane in three-dimensional space can be written as:

$$ax + by + c = z$$

where $a$, $b$, and $c$ are the plane parameters to be determined.

To apply the least squares method, we aim to minimize the sum of the squared distances between the observed points and the plane. The objective function to minimize is:

$$S(a, b, c) = \sum_{i=1}^{n} (ax_i + by_i + c - z_i)^2$$

where $n$ is the number of data points. These equations can be written in matrix form as:

$$\begin{bmatrix} x_0 & y_0 & 1 \\ x_1 & y_1 & 1 \\ \vdots & \vdots & \vdots \\ x_n & y_n & 1 \end{bmatrix} \begin{bmatrix} a \\ b \\ c \end{bmatrix} = \begin{bmatrix} z_0 \\ z_1 \\ \vdots \\ z_n \end{bmatrix} \tag{8}$$

Denoting the matrix on the left as $A$ and the vector on the right as $B$, we obtain a system of linear equations: $A\mathbf{w} = B$, where $\mathbf{w} = \begin{pmatrix} a \\ b \\ c \end{pmatrix}$.

The least squares solution is given by:

$$\mathbf{w} = (A^T A)^{-1} A^T B$$

This provides the optimal parameters $a$, $b$, and $c$ for the plane equation $ax + by + c = z$. Assuming the $z$-axis is aligned with gravity, the plane equation can be initially defined as $c = z$, where $x$ and $y$ are zero. Here, $c$ represents the average $z$-coordinate value of the point cloud.

In images where the floor or ceiling is not pre-estimated, the plane is defined as a surface parallel to another, separated by a predefined distance.

### A.3   Color bars for segmentation labels.

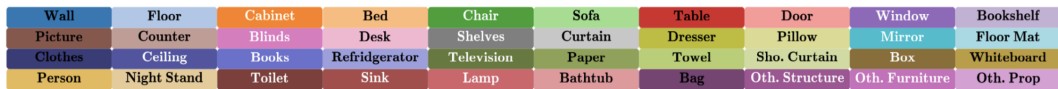

Figure 9: Stanford2D3DS dataset label colors.

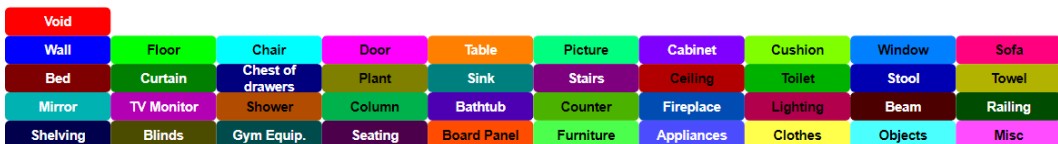

Figure 10: Structured3D dataset label colors.

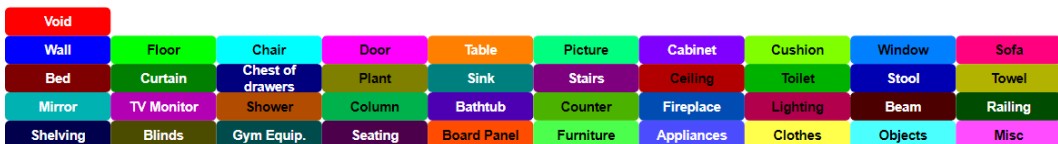

Figure 11: Matterport3D dataset label colors.

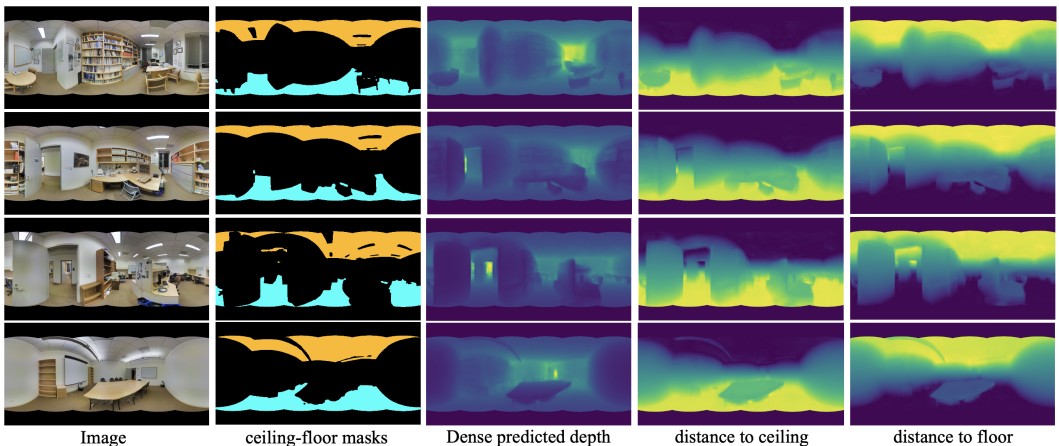

Figure 12: More examples of *distance to ceiling* and *distance to floor* masks, where light to dark colors represent distances from far to near.

**A.5** **2D/3D visualization of our approach step by step.**

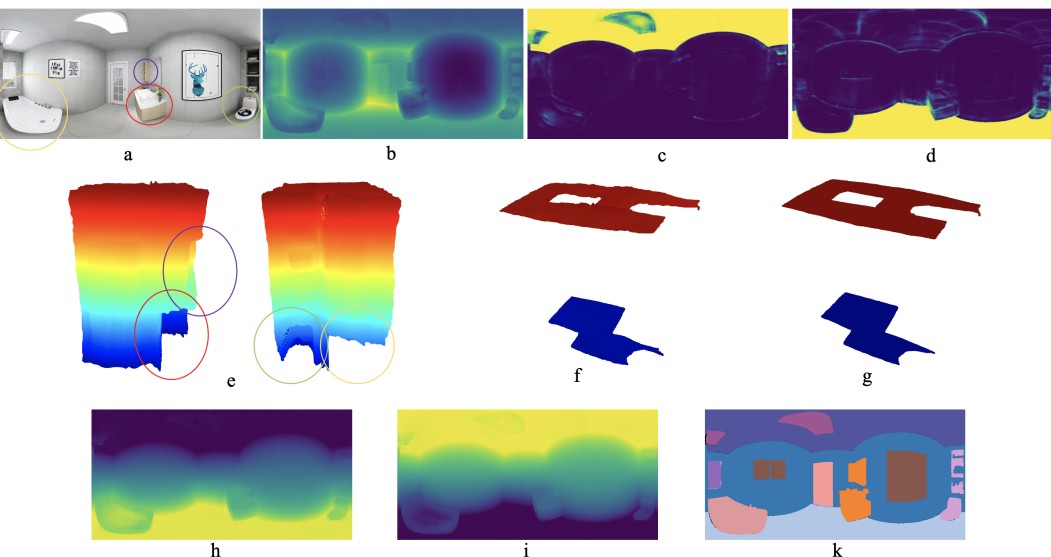

Figure 13: a) Input image. b) Predicted depth. c) Ceiling mask before softmax. d) Floor mask before softmax. e) Different views of pointcloud constructed from predicted depth. f) Ceiling and floor in 3D visualization. g) Planes of ceiling and floor in 3D coordinates after applying least square method. h) Distance of 3D points to ceiling plane. i) Distance of 3D points to floor plane. k) Final segmentation.

