# OpenReview forum: "Geometric Exploitation for Indoor Panoramic Semantic Segmentation"
_NeurIPS.cc/2024/Conference — NeurIPS 2024 poster_

### Official Review · Reviewer_DACe · 2024-07-12

**Soundness:** 3
**Presentation:** 2
**Contribution:** 2
**Rating:** 5
**Confidence:** 4

**Summary:**

The goal of the paper is panoramic semantic segmentation.  Given a panoramic image, produce a panoptic segmentation.

Main ideas:
1) Separate panoramic image into two sets of segments:
  over-sampled, representing planar objects such as ceiling and floor, and
  under-sampled, representing for other elements
2) Tailor optimization to expected properties of two sets:
  For oversampled:
   joint semantic segmentation and dense depth estimation
  For undersampled:
    add hand-crafted feature representing vertical relative distances.
  For combining:
    Transformer-based Context Module

Specific contributions:
1) separation of ceiling and floor from rest of scene,
2) vertical relative distances features
3) transformer-based context module

These contributions are different from, but conceptually related to, past work as follows:
1) The separation of ceiling and floor seems related in concept to previous work that separates the ground in lidar scans
2) The vertical relative distances features seems related to the HHA geocentric coordinate frame in "Perceptual Organization and Recognition of Indoor Scenes from RGB-D Images" (Gupta et al. 2013).

**Strengths:**

The paper describes a practical method for producing a semantic segmentation of panoramic images in indoor environments.  It proposes a few ideas motivated by the specifics of indoor scenes that improve the results.  These ideas are somewhat related to ones proposed in previous papers, and so they are not revolutionary -- however, the experiments show they are helpful in the tested setting.

**Weaknesses:**

It seems that there is an error in the calculation of the improvement provided by the method in table 1: SGAST4PASS gets 55.5 mIoU, and Trans4PASS++ gets 53.7 -- the difference is 1.8, not 2.8 as reported in the table (bottom row) and text (line 289).   This should be corrected.

It is not clear why the authors chose Trans4PASS++ as the baseline for this study.   There are newer methods that get better results (e.g., SFSS-MMSI, which appears in table 2).   In particular, it seems weird to have a table where one method performs a little bettern in val mIoU that the proposed one, and then list a +5.18 gap versus a midling method next to the result in the bottom row of the table.   Shouldn't the baseline be the current SOTA method?

It would have been nice for the authors to show results on Matterport3D, which has a wide variety of buildings with panoramic images and semantic segmentations (see the SFSS-MMSI paper).   S2D3D has only 13 classes, and all scenes are from similar buildings with low ceilings, and so the benchmark is not as varied and difficult.

The related work in section 2 lists and describes previous papers, but it does not describe their shortcomings or contrast them with the ideas in the submission.  As a result, the authors do not help the reader place the submission in the context of how it addresses an unsolved research problem.   I think it would be much better if the last sentence or two of each paragraph in section 2 lists why the listed methods do not solve the PASS problem, and how those limitations related to the approach taking in this paper.

**Questions:**

I understand the motivation for separating the ceiling and floor from the rest of the scene, partly because they are oversampled and partly because they are planar.   Does the same logic apply for walls?   I understand that they will not be as oversampled as the ceiling and floor in a panoramic image, but still they will be oversampled.   Plus, they are planar.   Did you consider separating them too?   If not, why not?   If so, why didn't it work?

**Limitations:**

Limitations are discussed in one sentence in the conclusion.   There is no discussion of handling more complex indoor environments, for example like a hotel lobby, or spa, or church, or ... where ceilings and floors are not so well identified.

---

> ### Author Rebuttal · Authors · 2024-08-06
>
> Dear Reviewer DACe, Thank you for appreciating our approach. We will address your comments below:
>
> **About the reviewer's summary**
>
> We would like to clarify that our work is about semantic segmentation of indoor panoramas, not panoptic segmentation as stated in the reviewer's summary. We also provide arguments as to why our approach is novel in indoor PASS:
>
> - Conceptual: To the best of our knowledge, the division of segments into over-sampled and under-sampled segments then optimising them based on geometric properties in our work is the first study in PASS, it directly exploits the unique properties that appear only in panoramic images(over-sampled segments are planar regions). Compare to 360x180 degree view in panoramic image, pinhole camera or normal lidar, images (depth images with lidar) cover only a part of the ceiling or the floor or neither of them, then the idea of division and segmentation with pinhole images is not feasible. So it should not be considered as a similar concept.
> - Furthermore, our motivation is not only a segments dividing method but also provide a tailor optimization strategy for each group. We analyze the unique constraint in the indoor scene that the total distance to the ceiling and floor surfaces tends to be a constant for each 3D point. Based on this observation, we develop 'vertical relative distance', a new representation that reflects the geometric relationship of 3D points to the constraint components of the scene. It can be seen as a new representation for indoor scene understanding.
>
> Considering the HHA geocentric coordinate frame in "Perceptual Organization and Recognition of Indoor Scenes from RGB-D Images". Given RGB-D data, the HHA geocentric frame is designed to estimate the direction of gravity, the algorithm tries to find the direction that is most aligned or most orthogonal to locally estimated surface normal directions at as many points as possible. Meanwhile, our approach segments and considers ceiling-floor as constraint components and then develops a distance-based concept for spatial representation. We did not find a closed similarity between them.
>
> Combine these arguments, we claim that our contribution are novel in PASS
>
> **Weaknesses:**
>
> **W1: Reason for choosing Trans4pass+ as our baseline**
>
> We verified that Trans4PASS+ already solved the distortion problem in PASS well and have become the baseline for recent methods, our goal is to divide large images into smaller groups of segments based on geometric properties to reduce the distortion gap between segments in each group and to tailor the learning strategy for each group. By combining our strategy with techniques already effectively proposed in TransPASS+, we aim to design a robust model for indoor PASS.
> SFSS-MMSI can be seen as the current SOTA before ours, but it is also based on Trans4PASS+, so is SGAT4PASS. Our goal is to develop a robust method which is also based on Trans4PASS+ to compete with these methods. It is a fact that when SFSS-MMSI was developed, SGAT4PASS was SOTA, but SFSS-MMSI is still based on Trans4PASS+ instead of SGAT4PASS.
>
> **W2: For the performance on Matterport3D dataset**
>
> Please refer the "Author Response to All Reviewers"
>
> **Question: Apply the same logic for the wall**
>
> We have already considered this approach, but there are some reasons why it should not be the same group as ceiling and floor. Firstly, **the semantic segment of the wall is not a plane**, unlike ceiling-floor, which normally appears as a single ceiling, single floor in each image, a single wall can be considered as a plane, but since it is a semantic segmentation, a class of multiple walls (parallel or orthogonal) cannot be considered as a plane (note that Plane Surface Normal Loss is only applied to planes), if the study is instance segmentation, it could also be separated. Secondly, the unique properties of the indoor scene is that the ceiling-floor sets the 3D upper-lower boundary, it could be considered as the constraint components of the room, then it facilitates the representation of spatial relationships of 3D points to constraints of the scene. It is true that the wall also appears in almost all images, but the appearance of the wall does not follow any rule, so adding the wall to this group does not make theoretical sense. Finally, adding the wall to the first group creates an imbalance between two groups, e.g. considering Figure 6 in the paper, ceiling-floor-wall may make up 75% of the image, while ceiling-floor may make up 40-50% of the image, depending on the dataset.
>
> **Limitation: For handling more complex indoor environments, for example like a hotel lobby, or spa**
>
> We agree with the reviewer that there are some special cases where ceilings and floors are not so well identified, in which case we define a virtual plane, for example: if no ceiling is identified in over-sampled segments, the ceiling plane will be a plane parallel to the floor, and the distance from the virtual plane to the floor will be the mean ceiling-floor distance analysed over the dataset. If there is no ceiling or floor in over-sampled segments, two virtual planes could be defined, but this is rare in indoor panoramas.

---

> > ### Comment · Reviewer_DACe · 2024-08-12
> > **Appreciate the authors' rebuttal**
> >
> > I appreciate all the work the authors put into the rebuttal.   The results on Matterport3D are particularly appreciated.   I also appreciate the new experiments aimed at teasing apart the effect of depth input/estimation on the results.
> >
> > I am willing to raise my review to borderline accept (i.e., "not going to argue if others want to accept").  Yet, I do not feel inclined to advocate strongly for acceptance myself because the core method (separating floor and ceiling and handling them separately) does not seem like a big conceptual jump, since people have done things like it for years -- especially separating out the ground from objects in outdoor environments.

---

> ### Author Response · Authors · 2024-08-08
> **Difference between vertical relative distance and HHA feature**
>
> Dear reviewer DACe,
>
> We assume that the mentioned concept is **Geocentric Pose: height above the ground** which was presented in the paper "Perceptual Organization and Recognition of Indoor Scenes from RGB-D Images" (Gupta et al. 2013)."
> We would like to provide further explanation to clarify the difference between our proposed concept **vertical relative distance** and the **height above the ground**:
>
> **Similarity:**
>
> Both "vertical relative distance" and "height above the ground" share the same intuition about measuring the distance between points in image to a constraint component of the images
>
> **Difference:**
> "height above the ground" take the 2D depth image as input and consider height above the lowest point in
> the 2D image as a surrogate for the height from the supporting ground plane. Since it works with 2D images, in case camera intrinsic is not use, it will sometimes be incorrect in reflecting the relative height between different points due to the ill-posed problem; if camera intrinsic is used, it can be treated as a point-to-point distance rather than a distance to the ground. Meanwhile, our "vertical relative distance" is modeled in 3D coordinates with pre-defined ceiling-floor, which facilitate the concept to reflect the relative distance from 3D points to the pre-defined plane accurately.
>
> In summary, both "vertical relative distance" and "height above the ground" share the motivation of distance-based concept, but because of different input, pre-defined component, we think that it can be considered as different representation.

---

> ### Author Response · Authors · 2024-08-13
>
> We thank the reviewer for the appreciation and reconsideration of our work, we understand the reviewer's concern as the similar motivation can be seen before, but for lidar scanning. During the rebuttal phase, we also provided some arguments to support why our idea can be seen as novel in indoor PASS, we believe that our reformulated concept for indoor PASS, as well as the distance-based method, will be beneficial to the community, especially the holistic understanding method for indoor scenes. Again, we thank the reviewer for the constructive feedback and discussion.

---

### Official Review · Reviewer_irP4 · 2024-07-15

**Soundness:** 3
**Presentation:** 3
**Contribution:** 3
**Rating:** 6
**Confidence:** 2

**Summary:**

The method divides the indoor panorama semantic segmentation problem into the prediction of over-sampled segmentation (like ceiling, floor, and planar objects) and under-sampled segmentation (like objects in indoor scenery like furniture, windows, door, etc.) subtasks.

The paper utilizes over-sampled segment prediction with multi-task semantic and depth estimation, to provide spatial relationships of the 3D scene objects concerning the planar constraints in the form of vertical relative distances. The method presents a transformer-based attention mechanism to aggregate the obtained geometric feature representations from the over-sampled segment prediction branch to estimate the under-sampled segments. The prediction from both subtasks is then merged to produce the final panorama semantic segmentation result. Their method produces better results than the current SOTA on the mentioned standard datasets.

**Strengths:**

Overall, the paper is well-written and easy to follow. The method provides a geometric representation called vertical relative distance of 3D scene points concerning the ceiling and floor planar context that provides the additional spatial relationship to better estimate the challenging objects in the indoor panorama scenery. The dual branch network provides slightly better performance than the methods listed in the paper

**Weaknesses:**

•	The current method is computationally complex compared to the current methods.
•	The process produces unknown segments due to the merging of the two separate groups of segment predictions.
•	The performance improvement of the proposed heavy capacity model does not seem significant
•	The ablation study of the effect of different utilized geometric representations might require qualitative comparisons highlighting the effect of each module as the performance improvement seems limited
•	Lack of reproducibility

**Questions:**

•	Network details of deformable MLP, segmentation, and depth head module need to be discussed.
•	Detailed loss function equation missing.
•	Line 125: HoHoNet [22] constructed a framework ‘for of’ layout structure joint per-pixel dense prediction tasks, e.g., depth estimation, and semantic segmentation based on features of 1D horizontal representation seems mistaken.

•	The paper misses mentioning the concept and/or gaps of the current compared methods called tangent [10], SFSS [12] and Panoformer [20] in the related work.

**Limitations:**

Overall, the idea of dividing the segmentation task into planar and object segmentation subtasks while utilizing easily obtained planar geometric representation to provide spatial relationships to help segment challenging objects seems distinctive but seems to show limited performance improvement using the heavy capacity model. Also, the paper misses the mentioned module, and related work details which need to be addressed.

---

> ### Author Rebuttal · Authors · 2024-08-07
>
> Dear Reviewer irP4, Thank you for appreciating our approach. We will address your comments below.
>
> **Problem 1: About the heavy computationally complexity**
>
> In fact, our model is slightly heavier than the baseline (Trans4PASS+) with 53M and 39M parameters respectively. However, in terms of TFLOPS, it is approximately the same as the baseline, which means that the execution time of our model is the same as the baseline. Since the high computational complexity is mainly caused by the Transformer-based Context Module, it can be solved by finding the most appropriate number of transformer layers for each dataset.
>
> **Problem 2: About the limit improvement performance.**
>
> Overall, our method outperforms the baseline (Trans4PASS+) with a large improvement. Besides, our model also provide the better performance compared to the SOTA SFSS-MMSI except for validation on Structured3D dataset, we confirm that this is just test setup problem. SFSS-MMSI choose the best model on val-set to perform on test, while we conducted the opposite way, if we follow the same test setup, our method surpasses SFSS-MMSI on both val and test sets with 72.86% (val) and 71.66% (test), respectively. We also mention the comparison on Matterport3D dataset, please refer the "Author Response to All Reviewers".
>
> **Problem 3: For the writing problem**
>
> Thank the reviewer for pointing out the writing problem, the lack of mention of the concept, and the incomplete related work, it could be addressed in the next version.

---

### Official Review · Reviewer_GGrZ · 2024-07-20

**Soundness:** 3
**Presentation:** 3
**Contribution:** 3
**Rating:** 5
**Confidence:** 5

**Summary:**

This paper introduces a novel approach to panoramic semantic segmentation. The work views panoramic segmentation from two perspectives including over-sampled segmentation and under-sampled segmentation. The rich geometric depth information is exploited using a transformer-driven context module. The experiments on two public datasets demonstrate the effectiveness of the proposed model.

**Strengths:**

1. The idea of studying over-sampled and under-sampled segments in panoramic segmentation is interesting.
2. The paper is overall well-written and nicely structured.

**Weaknesses:**

1. The section 3.5 should be better formalized to help the readers understand the merging process.
2. The proposed framework contains a lot of steps, which could lead to increased running time. It would be nice if the authors could present the computation complexity of different steps in the framework.
3.  Matterport3D is also an important benchmark of panoramic indoor segmentation. It would be nice to evaluate the proposed method on the Matterport3D as well.
4. Regarding the vertical relative distance, would you directly compare the proposed representation against more recent state-of-the-art representations or distance measures to verify the superiority of your proposed method?
5. The proposed transformer-based context module should be compared against existing state-of-the-art decoders and context aggregation methods like ASPP, PSP, SegFormer, UperNet, FPN, Mask2Former, etc.
6. Recent panoramic segmentation methods like MultiPanoWise and DATR could be discussed and compared.

**Questions:**

1. How about the current occlusion-aware seamless segmentation, which is relevant for panoramic scene understanding?
2. Following SFSS-MMSI, would you consider providing more multi-modal ablation results for analysis? For example, when using different representations, how would the performance change?
3. Would you consider visualizing some feature maps or attention maps to help better understand the effectiveness of the proposed method?
4. Is it possible to evaluate on outdoor panoramic segmentation benchmarks? For example, some outdoor panorama datasets could be enriched with geometric information by using large models like depth anything. This could be discussed.

**Limitations:**

This paper has clearly discussed the limitations.

---

> ### Author Rebuttal · Authors · 2024-08-05
>
> Dear Reviewer GGrZ, Thank you for appreciating our approach. We will address your comments (both weaknesses and questions) below.
>
> **W1:** About merging process, we agree with the reviewer that the merging processed should be described and visualized clearly for better understanding
>
> **W2:** About computation complexity, we provide the measurement on main components, it can be referred as table below:
>
> | Module    |  Params (G)  |  TFLOPs |
> |----------|:-------------:|------:|
> | Encoder |  0.024 | 0.058 |
> | Over-sampled segments joint depth estimation|   0.003 |0.010 |
> | Transformer-based Context Module related |  0.024  |0.061|
> | Under-sampled segments |  0.002   |  0.006 |
> | Total |  0.053   |  0.135 |
>
> **W3:**  About the performance on Matterport3D dataset, please refer the "Author Response to All Reviewers"
>
> **W4:** Comparison with SOTA distance based representation
>
> Our work: dividing segments into sub-groups, tailor optimization with different strategy, and setting constraint components to form the concept of vertical relative distance are novel approaches in the PASS, we did not find any same concept in the previous works to compare, instead of that, we conduct experiment to verify the effectiveness of proposed concept as following below:
>
> 1) Model performance with and without vertical relative distance as input of Transformer based Context Module (table 4 of main paper)
> 2) Beside vertical relative distance, we also considered another concept, which measures the angle between the normal vector of the ceiling-floor and the normal vector of the 3D points constructed by the predicted depth, but after adding it as an input to the Transformer-based Context Module, the performance did not change significantly. The reason for the ineffectiveness of this concept can be explained as follows: unlike the distance-based representation, where the vertical relative distance of two adjacent points in 3D is quite similar, the point-level normal vector estimated from depth is really sensitive with noise, making it more difficult for the model to achieve coverage.
>
> **W5:** Compare to other context aggregation techniques
>
> We present the quantitative results on the Stanford dataset with different selections of context aggregation techniques, details of this comparison can be seen:
> | Module    |  mIoU(%)  |
> |----------|:-------------:|
> | Segformer decoder|  53.4 |
> | ASPP |  54.6  |
> | Transformer-based Context Module | **56.8**   |
>
> We realised that the SegFormer decoder is lightweight but performs poorly in our model, and that ASPP is not efficient for the context aggregation task.
>
> **W6:** we agree that some models such as MultiPanoWise and DATR could be discussed and compared
>
> **Q1: About occlusion-aware seamless segmentation**
>
> We assume that the mentioned paper is "Occlusion-aware seamless segmentation (OASS) - ECCV24". As the paper was published after our research, we did not consider this paper in our study. As we know, OASS has been introduced as a new task in panoramic semantic segmentation, this paper aims to solve the unsupervised domain adaptation between pinhole and panoramic outdoor image by presenting two related techniques, unmasking mechanism to solve the object occlusions and modification of Deformable Patch Embedding to reduce the image distortion. The objective and approach of this paper is different from our work.
>
> **Q2: Analysis of multi-modal ablation results**
>
> Thank reviewer for the recommendation, combining our idea with multi representation as input as SFSS-MMSI is interesting and this combination will be conducted
>
> **Q3: Visualization of feature map or attention map**
>
> For better understanding of each step in our pipeline, we provide the visualization in the attached pdf file at the "Author Response to All Reviewers"
>
> **Q4: Perform on outdoor dataset**
>
> Using this approach for outdoor panoramic segmentation is worth considering, since the road can be considered as a planar object and the superiority of the depth anything model also facilitates this approach, but currently there are some limitations that limit the applicability of our approach in outdoor scenes
> 1) Lack of labels for outdoor panoramic depth datasets: As described in the paper, our framework requires dense depth annotation and semantic segmentation for supervised learning. These requirements can be adapted for indoor datasets such as Stanford, Structured3D and Matterport3D. Considering an outdoor dataset for a panoramic semantic segmentation task, Cityscapes and SynPASS provide 5000 and 9080 high quality dense pixel annotations respectively, but lack dense depth annotations. In this case, the use of a large pre-trained model such as depth anything should be a possible approach, but this approach limits the joint learning and increases the runtime of the whole framework due to the execution of a large model such as depth anything. In addition, since the depth anything model is optimized for pinhole images, domain adaptation is also required to be applicable to PASS. In fact, PASS for outdoor is typically solved by unsupervised domain adaptation as presented by Trans4PASS+ or OASS. Meanwhile, the DensePASS dataset provides only 100 images with segmentation annotations, which are intended for unsupervised domain adaptation testing.
> 2) For the indoor scene, the depth range is quite small, so the relative depth estimation is quite correct, which facilitates the "vertical relative distance" to accurately reflect the spartial properties, for example, the distance from the ceiling to a chair and that to the table are distinguishable . In contrast, for the outdoor scene, with large depth scales, the imperfect depth estimation limits the impact of the distance-based concept. for example, the "vertical relative distance" of some segments in the outdoor scene is unclear, e.g. the vertical gap between the road-sidewalk, road-road markings are difficult to distinguish.

---

> > ### Comment · Reviewer_GGrZ · 2024-08-08
> > **Comment**
> >
> > The rebuttal helps to solve many of the concerns. The improvement of the proposed model over the baseline is significant. The direct comparison of the Transformer-based Context Module against existing decoders shows the gains of the proposed method.
> >
> > The reviewer would like to elevate the rating of borderline acceptance. The reviewer suggests that the benefit of the proposed method for tackling distortions could be better illustrated in the results of qualitative visualization. Besides, the gap between indoor PASS (Supervised, Multimodal) and outdoor PASS (UDA) would be worth discussing in the paper.
> >
> > Sincerely,

---

> > > ### Author Response · Authors · 2024-08-13
> > >
> > > We thank the reviewer for the appreciation and reconsideration of our work, we agree with the reviewer that the impact of the segment dividing strategy for reducing distortions across segments on each group should be illustrated in the qualitative visualization. Furthermore, the, the gap between indoor PASS (Supervised, Multimodal) and outdoor PASS (UDA) could be discussed in the revised version. Again, we thank the reviewer for the constructive feedback and discussion.

---

### Official Review · Reviewer_QfCk · 2024-07-30

**Soundness:** 3
**Presentation:** 3
**Contribution:** 3
**Rating:** 5
**Confidence:** 4

**Summary:**

The authors decompose the indoor panoramic semantic segmentation task into two sub-tasks: segmentation and depth estimation and design to enhance the geometric information. Specifically, the method first introduces the vertical relative distance to demonstrate the relationships between planar objects (ceiling and floor) and other object pixels in 3D coordinates. Then it aggregates various representations into a transformer-based context module to learn the geometric context. The experimental results partially prove the efficiency of the method proposed by the authors.

**Strengths:**

1.	The authors consider segmenting the planar objects (which make up about 40% of a panoramic image) and other objects in separate strategies, which alleviate the negative impact caused by the imbalance category distribution.
2.	The authors consider implicitly modeling the spatial relationships by defining vertical relative distance, which is adaptive to the indoor scenes.

**Weaknesses:**

1.	The authors claimed in Line 54 that the method is proposed to deal with the various distortions present in panoramic images. However, the proposed method improved the performance mainly by incorporating more representations rather than aiming for distortions, and the distortion problem is not highlighted in the qualitative results.
2.	Although the input of the proposed model is only RGB images, the method utilizes the depth ground truth for the supervision of the depth estimation task, thus the comparisons with the state-of-the-art methods that utilize only RGB in Table 2 are unfair.
3.	The ablation studies without the depth estimation task are needed, for instance, the authors can supplement in Table 4 the experimental results of “F_img+F_h+F_m”.
4.	The qualitative comparisons and analysis without the depth estimation task are suggested to supplement in the main paper.
5.	It is suggested to conduct on more datasets such as Matterport3D to verify the general ability of the proposed method.
6.	Minor: The DPE in Figure 2 is suggested to specify.

**Questions:**

Please the weakness section.

**Limitations:**

The authors mentioned the limitations in the conclusion section.

---

> ### Author Rebuttal · Authors · 2024-08-05
>
> Dear Reviewer QfCk,
> Thank you for appreciating our approach. We will address your comments below.
>
> **Q1: About the wrong claim**
>
> We agree with the reviewer that this sentence should be changed, our meaning is 'to reduce the distortion gap between segments in each group'.
>
> **Q2: About the unfair comparison**
>
> Since our work utilizes the depth ground truth for the supervision of the depth estimation task, thus the comparisons with SOTA methods only RGB may be unfair. Actually, in the table 2, we compare our framework with two kinds of previous work:
> 1) Input RGB and output semantic segmentation: SGAT4PASS, Trans4PASS, Trans4PASS+, SFSS-MMSI(RGB)
> 2) Input RGB and output depth estimation & semantic segmentation as ours: FreDSNet
>
> We also mention the quantitative result of SFSS-MMSI (input RGB+depth and output semantic segmentation), please note that comparing our method with PanoFormer and SFSS-MMSI, which inputs both RGB and ground truth depth, is also unfair. With ground truth depth as input, the PanoFormer or SFSS-MMSI can exploit the geometry more accurately than depth by learning as our method. Please refer the "Author Response to All Reviewers" for more detail.
>
> **Q3+Q4: Performance of model without the involvement of depth**
>
> Thank the reviewer for the recommendation. Due to time constraints, we just additionally considered ablation studies on the Stanford dataset with two settings. First, we keep the segment partitioning and optimization strategy and remove the depth branch as well as F_pc, F_dist from the input of the Transformer-based Context Module. Second, we still keep the joint learning over-sampled segments with depth estimation, but remove F_pc and F_dist from the Transformer-based Context Module, detail of the quantitative comparison is shown below:
>
> | Methods   |      Avg mIoU (%)      |   F1 mIoU (%) |
> |----------|:-------------:|------:|
> | baseline |  53.7 | 53.6 |
> | w/o depth involvement (both depth estimation and F_pc, F_dist) |    54.3 |   54.7 |
> | with depth estimation but no F_pc and F_dist|    54.6 |   55.0 |
> | with depth involvement (depth estimation and F_pc, F_dist) |  **55.5**  |  **56.8**|
>
> It can be seen that without depth involvement (both depth estimation and F_pc, F_dist) the performance is still slightly higher than the baseline due to the effectiveness of the proposed segments dividing strategy.  In case of including depth estimation but without F_pc and F_dist, the performance increase is mainly for ceiling-floor, then Avg mIoU just increases slightly. With the full involvement, the model become more robust, indicated by significant improvement with mentioned settings.
> As suggested by the reviewer, more ablation studies should be conducted and mentioned in the main paper.
>
> **Q5: Performance on Matterport3D dataset**
> Please refer the "Author Response to All Reviewers"
>
> **Q6: Specify DPE**
> In fact, the specification of DPE was introduced in the Trans4PASS+ paper, and should be mentioned in more detail in the paper.

---

> > ### Comment · Reviewer_QfCk · 2024-08-14
> >
> > Thanks for the efforts on the experiments on the ablation study of the use of depth and Matterport3D dataset. Nonethless, I have the same concern on the contribution with Reviewer DACe. Therefore, I raise my rating to borderline accept.

---

> ### Author Response · Authors · 2024-08-14
>
> Dear reviewer QfCk,
>
> We sincerely thank the reviewer for the discussion. Regarding the concern as the reviewer DACe, we have provided the explanation to address and claim our novelty, it will be more clarified in the revised version.

---

### Author Rebuttal · Authors · 2024-08-07

Dear all reviewers:

We sincerely appreciate the reviewers for the time and efforts on the review. We first address some common questions, followed by detailed responses to each reviewer separately. We hope our responses clarify existing doubts. We will really appreciate it if reviewer QfCk, GGrZ and DACe can kindly reconsider the decision, because of the novelty and good performance of our approach as well as the main comments are well addressed.

**Update the experiment on the Matterport3D dataset.**

Initially, we dicided to do not conduct the evaluation on the Matterport3D dataset for two reasons: first, the comparison on the Matterport3D dataset did not appear in the recent methodologies (Trans4PASS, Trans4PASS+, SGAT4PASS, FreDSNet), second, the quantitative result of SOTA on the Matterport3D dataset is quite limited, it raises skeptic about the quality of this dataset. However, following the recommendation of almost all reviewers, we update the performance(mIoU%) on this dataset and then update the full comparison table as follows:

| Methods    |Input|  Stanford - fold1  |Stanford - avg|  Structured3D-val |Structured3D-test|  Matterport3D - val|  Matterport3D - test|
|----------|:------:|------:|------:|------:|------:|------:|------:|
| PanoFormer |RGB|  - | 52.35|55.57 |54.87|30.04|26.87|
| Trans4PASS+ |RGB|  53.60 | 53.70|66.74 |66.90|33.43|29.11|
| SFSS-MMSI|RGB|  53.43 |52.87 |71.94|68.34|35.15|31.30|
| PanoFormer |RGB+Depth|  - | **57.03**|60.98 |59.27|33.99|31.23|
| SFSS-MMSI|RGB+Depth|55.98| 55.49  |**73.78**|  70.17| **39.19**|**35.92**|
| Ours |RGB| 56.80| 55.50 |72.86|**71.66**|36.42|33.06|

There are two notes in this table.
1) In the main paper, we reported the performance of our in Structured3D val and test are: 71.92 and 71.70, but we just realized that SFSS-MMSI chose the best checkpoint on val set then perform on the test set, while we did the opposite way, after following the existing method, we correct the quantitative results as shown in this table.
2) When the input is RGB images, our method outperforms the baseline as well as the SOTA method in all test datasets.
In addition, we also report the performance of the PanoFormer and SFSS-MMSI with the RGB and depth ground truth as input. We would like to comment that this comparison between our method (input: RGB) with PanoFormer and SFSS-MMSI (input: RGB+depth) is unfair, with corrected depth values as input, PanoFormer or SFSS-MMSI has more advantages to understand the corrected geometry than estimating depth by learning like ours. However, our approach still proves the robustness, gives a better result on the Structured3D test set, and is competitive on other datasets. Once again, these results show the effectiveness of our proposed method.

**Visualisation for a better understanding of our approach**

As requested by reviewer **GGrZ**, we provide the visualisation for better understanding, please check the pdf file for content

**Other concerns**.

We thank all reviewers for helping us to find out some errors related to writing, tables, ... All these errors should be treated carefully.

---

### Author Response · Authors · 2024-08-14
**Looking for the responses from the reviewers**

Dear Area Chairs and Reviewers,

There are 10 hours left until the end of discussion phase, but we still did not get the the responses from all reviewers even though we solved many concerns in the rebuttal phase, this comment is to express that we really appreciate if the responses is updated, it will make the fair for our paper evaluation.

Sincerely

---

> ### Author Response · Authors · 2024-08-14
>
> Dear all reviewers,
>
> We would like to thank all the reviewers for the valuable feedback and recommendations during the discussion phase.
>
> Sincerely

---

### Decision · Program_Chairs · 2024-09-25

**Decision:**

Accept (poster)

**Comment:**

After the discussion phase all reviewers lean toward acceptance, noting that the manuscript is well written and has novelty. Concerns raised were primarily about the evaluation (e.g., requesting additional ablations, performance on a missing benchmark, study of computational complexity). A detailed rebuttal was submitted that helped to address many of the reviewer concerns, including adding comparisons on Matterport3D and an ablation study related to depth estimation. After the reviewer-author discussion phase, three of the four reviewers raised their rating to borderline accept. As all reviewers are in agreement, and the authors demonstrated commitment to improving the manuscript in the rebuttal phase, the AC reached a decision to accept the paper.  Please take the reviewer feedback into account when preparing the camera ready version.